# Benchmarking of eight recurrent neural network variants for breath phase and adventitious sound detection on a self-developed open-access lung sound database—HF_Lung_V1

**Fu-Shun Hsu[1,2,3], Shang-Ran Huang [3], Chien-Wen Huang[4], Chao-Jung Huang[5], Yuan-Ren Cheng[3,6,7], Chun-Chieh Chen[4], Jack Hsiao[8], Chung-Wei Chen[2], Li-Chin Chen[9], Yen-Chun Lai[3], Bi-Fang Hsu[3], Nian-Jhen Lin[3,10], Wan-Ling Tsai [3], Yi-Lin Wu[3], Tzu-Ling Tseng[3], Ching-Ting Tseng[3], Yi-Tsun Chen[3], Feipei Lai[1] ***

**1** Graduate Institute of Biomedical Electronics and Bioinformatics, National Taiwan University, Taipei, Taiwan, **2** Department of Critical Care Medicine, Far Eastern Memorial Hospital, New Taipei, Taiwan, **3** Heroic Faith Medical Science Co., Ltd., Taipei, Taiwan, **4** Avalanche Computing Inc., Taipei, Taiwan, **5** Joint Research Center for Artificial Intelligence Technology and All Vista Healthcare, National Taiwan University, Taipei, Taiwan, **6** Department of Life Science, College of Life Science, National Taiwan University, Taipei, Taiwan, **7** Institute of Biomedical Sciences, Academia Sinica, Taipei, Taiwan, **8** HCC Healthcare Group, New Taipei, Taiwan, **9** Research Center for Information Technology Innovation, Academia Sinica, Taipei, Taiwan, **10** Division of Pulmonary Medicine, Far Eastern Memorial Hospital, New Taipei, Taiwan

* flai@csie.ntu.edu.tw

## Abstract

A reliable, remote, and continuous real-time respiratory sound monitor with automated respiratory sound analysis ability is urgently required in many clinical scenarios—such as in monitoring disease progression of coronavirus disease 2019—to replace conventional auscultation with a handheld stethoscope. However, a robust computerized respiratory sound analysis algorithm for breath phase detection and adventitious sound detection at the recording level has not yet been validated in practical applications. In this study, we developed a lung sound database (HF_Lung_V1) comprising 9,765 audio files of lung sounds (duration of 15 s each), 34,095 inhalation labels, 18,349 exhalation labels, 13,883 continuous adventitious sound (CAS) labels (comprising 8,457 wheeze labels, 686 stridor labels, and 4,740 rhonchus labels), and 15,606 discontinuous adventitious sound labels (all crackles). We conducted benchmark tests using long short-term memory (LSTM), gated recurrent unit (GRU), bidirectional LSTM (BiLSTM), bidirectional GRU (BiGRU), convolutional neural network (CNN)-LSTM, CNN-GRU, CNN-BiLSTM, and CNN-BiGRU models for breath phase detection and adventitious sound detection. We also conducted a performance comparison between the LSTM-based and GRU-based models, between unidirectional and bidirectional models, and between models with and without a CNN. The results revealed that these models exhibited adequate performance in lung sound analysis. The GRU-based models outperformed, in terms of *F1* scores and areas under the receiver operating characteristic curves, the LSTM-based models in most of the defined tasks. Furthermore, all

**Data Availability Statement:** All relevant data are within the manuscript and its Supporting Information files.

**Funding:** Raising Children Medical Foundation, Taiwan, fully funded all of the lung sound collection and contributed the recordings to Taiwan Society of Emergency and Critical Care Medicine. The Heroic Faith Medical Science Co. Ltd, Taipei, Taiwan, freely provided the lung sound recording device (HF-Type-1) for the study and fully sponsored the data labeling and deep learning model training. There was no additional external funding received for this study. The funders had no role in study design, data collection and analysis, decision to publish, or preparation of the manuscript.

**Competing interests:** FSH, SRH, YRC, YCL, BFH, YLW, TLT and CTT are full-time employees and CJH, NJL, WLT and YTC are part-time employees of Heroic Faith Medical Science Co. Ltd. CWH and CHC are with Avalanche Computing Inc., whom Heroic Faith Medical Science Co. Ltd. commissioned to train the deep learning models. This does not alter our adherence to PLOS ONE policies on sharing data and materials."

bidirectional models outperformed their unidirectional counterparts. Finally, the addition of a CNN improved the accuracy of lung sound analysis, especially in the CAS detection tasks.

## Introduction

Respiration is vital for the normal functioning of the human body. Therefore, clinical physicians are frequently required to examine respiratory conditions. Respiratory auscultation [1–3] using a stethoscope has long been a crucial first-line physical examination. The chestpiece of a stethoscope is usually placed on a patient's chest or back for lung sound auscultation or over the patient's tracheal region for tracheal sound auscultation. During auscultation, breath cycles can be inferred, which help clinical physicians evaluate the patient's respiratory rate. In addition, pulmonary pathologies are suspected when the frequency or intensity of respiratory sounds changes or when adventitious sounds, including continuous adventitious sounds (CASs) and discontinuous adventitious sounds (DASs), are identified [1, 2, 4]. However, auscultation performed using a conventional handheld stethoscope involves some limitations [4, 5]. First, the interpretation of auscultation results substantially depends on the subjectivity of the practitioners. Even experienced clinicians might not have high consensus rates in their interpretations of auscultatory manifestations [6, 7]. Second, auscultation is a qualitative analysis method. Comparing auscultation results between individuals and quantifying the sound change by reviewing historical records are difficult tasks. Third, prolonged continuous monitoring of lung sound is almost impractical. Lastly, a practitioner wearing personal protective equipment finds it difficult to perform auscultation without breaching the protection [8], which limits the use of auscultation on a patient with an airborne or droplet-transmitted pulmonary disease, such as coronavirus disease 2019 (COVID-19) [9–11]. To overcome the aforementioned limitations, computerized respiratory sound analysis [12] is required. Furthermore, a tele-auscultation system [13, 14] with automated respiratory sound analysis can be realized in the form of a mobile app or web service with proper infrastructure supports, which can facilitate remote respiratory monitoring not only in a clinical setting but in a home-care setting. The tele-auscultation system can greatly help during care of COVID-19 patients.

In the past, traditional machine learning methods were commonly used to build a computerized analytical model for respiratory sound analysis [4, 12, 15, 16]. However, it is believed that the improvement of the performance of traditional machine learning models may hit a plateau as the amount of data increases beyond a certain number; however, deep learning models do not have such concerns [17]. Moreover, deep learning approach is highly scalable for dealing with the problems with different complexity [17]. Lastly, deep learning models can learn the optimal features and do not rely on handcrafted feature engineering which depends on domain knowledge [17, 18]. Therefore, more and more researchers turn their attention to using deep learning in recent years. Besides, many studies have used deep learning to combat the COVID-19 outbreak [19] because the problems may involve high complexity and large amounts of data are rapidly generated and aggregated from all parts of the world.

However, previously proposed deep learning models for respiratory sound analysis may have been limited by insufficient data. As of writing this paper, the largest reported open-accessed respiratory sound database was organized in a scientific challenge at the International Conference on Biomedical and Health Informatics (ICBHI) 2017 [20, 21], which comprises 6,898 breath cycles, 1898 wheezes and 8,877 crackles acquired from the lung sounds of 126 individuals. As data size plays a major role in the creation of a robust and accurate deep learning-based algorithm [22, 23], a larger open-access dataset could further benefit the development of more accurate respiratory sound analysis models.

In addition, computerized respiratory sound analysis can be categorized into different tasks at different levels, namely sound classification at the segment, event and recording levels and sound detection at the segment and recording levels [4]. To our best knowledge, almost all previous studies have focused on only distinguishing healthy participants from participants with respiratory disorders and classifying normal breathing sounds and various types of adventitious sounds. Only few studies have reported the performance of sound detection at the recording level using deep learning based on private datasets [24–26]. Accurate detection of the start and end times of breath phase and adventitious sounds can be used to derive quantitative indexes, such as duration and occupation rate, which are potential outcome measures for respiratory therapy [27, 28]. Therefore, it is worthwhile to pursue sound detection at the recording level in respiratory sound analysis.

Accordingly, the aims of the present study were to establish a large and open-access respiratory sound database for training algorithms for the detection of breath phase and adventitious sounds at the recording level, mainly focusing on lung sounds, and to conduct benchmark tests on the established database using deep neural networks. Recurrent neural networks (RNNs) [29] are effective for time-series analysis; long short-term memory (LSTM) [30] and gated recurrent unit (GRU) [31] networks, which are two RNN variants, exhibit superior performance to the original RNN model. However, whether LSTM models are superior to GRU models (and vice versa) in many applications, particularly in respiratory sound analysis, is inconclusive. Bidirectional RNN models [32, 33] can transfer not only past information to the future but also future information to the past; these models consistently exhibit superior performance to unidirectional RNN models in many applications [34–36] as well as in breath phase and crackle detection [25]. However, whether bidirectional RNN models outperform unidirectional RNN models in CAS detection has yet to be determined. Furthermore, the convolutional neural network (CNN)–RNN structure has been proven to be suitable for heart sound analysis [37], lung sound analysis [38], and other tasks [35, 39]. Nevertheless, the application of the CNN–RNN structure in respiratory sound detection has yet to be fully investigated. Hence, we chose to use eight RNN-based variants, namely, LSTM, GRU, bidirectional LSTM (BiLSTM), bidirectional GRU (BiGRU), CNN-LSTM, CNN-GRU, CNN-BiLSTM, and CNN-BiGRU models, for the benchmark tests. Benchmarking can demonstrate the reliability and goodness of a database and provide a baseline reference for the future studies. It can also be applied to investigate the performance of the RNN variants in respiratory analysis.

In summary, the aims of this study are outlined as follows:

- Establish the largest open-access lung sound database as of writing this paper—HF_Lung_V1 (https://gitlab.com/techsupportHF/HF_Lung_V1).

- Conduct benchmark tests of breath phase and adventitious sounds detection at the recording level using the eight aforementioned RNN-based models based on the lung sound data.

- Conduct a performance comparison between LSTM and GRU models, between unidirectional and bidirectional models, and between models with and without a CNN in breath phase and adventitious sound detection based on the lung sound data.

- Discuss factors influencing model performance.

## Related work

Most previous deep learning studies in lung sound analysis have focused on the classification of healthy participants and participants with respiratory diseases [40–46] and the classification

of normal breathing sounds and various types of adventitious sounds [38, 46–65]. The models in most of these studies are developed on the basis of an open-access ICBHI database [20, 21].

Only few previous studies have explored the use of deep learning for detecting the breath phase and adventitious sounds at the recording level. In our previous study, we developed an attention-based encoder-decoder model for breath phase detection and achieved an *F1* score of approximately 90% for inhalation detection and an *F1* score of approximately 93% for exhalation detection. However, the dataset used was relatively small (489 15-s recordings) and the task was to detect inspiratory and expiratory sounds in a 0.1-s segment (time frame) instead of detecting the events of inhalations and exhalations. Messner et al. [25] applied the BiGRU to two features, namely Mel-frequency cepstral coefficients (MFCCs) and short-time Fourier transform (STFT)-derived spectrograms, and achieved an *F1* score of approximately 86% for breath phase detection based on 4,656 inhalations and 4,720 exhalations and an *F1* score of approximately 72% for crackle detection based on 1,339 crackle events. Jácome et al. [26] used a faster region-based CNN (Faster R-CNN) framework to obtain a sensitivity of 97.5% and specificity of 85% in inspiratory phase detection and a sensitivity of 95.5% and specificity of 82.5% in expiratory phase detection, which was based on a dataset comprising 3,212 inspiratory phases and 2,842 expiratory phases. The datasets used in the three studies are not open to the public.

## Establishment of the lung sound database

### Data sources and patients

The lung sound database was established using two sources. The first source was a database used in a datathon in Taiwan Smart Emergency and Critical Care (TSECC), 2020, under the license of Creative Commons Attribution 4.0 (CC BY 4.0), provided by the Taiwan Society of Emergency and Critical Care Medicine. Lung sound recordings in the TSECC database were acquired from 261 patients.

The second source was sound recordings acquired from 18 residents of a respiratory care ward (RCW) or a respiratory care center (RCC) in Northern Taiwan between August 2018 and October 2019. The recordings were approved by the Research Ethics Review Committee of Far Eastern Memorial Hospital (case number: 107052-F). Written informed consent was obtained from the 18 patients. This study was conducted in accordance with the 1964 Helsinki Declaration and its later amendments or comparable ethical standards.

All patients were Taiwanese and aged older than 20 years. Descriptive statistics regarding the patients' demographic data, major diagnosis, and comorbidities are presented in Table 1; however, information on the patients in the TSECC database is missing. Moreover, all 18 RCW/RCC residents were under mechanical ventilation.

### Sound recording

Breathing lung sounds were recorded using two devices: (1) a commercial electronic stethoscope (Littmann 3200, 3M, Saint Paul, Minnesota, USA) and (2) a customized multichannel acoustic recording device (HF-Type-1) that supports the connection of eight electret microphones. The signals collected by the HF-Type-1 device were transmitted to a tablet (Surface Pro 6, Microsoft, Redmond, Washington, USA; Fig 1). Breathing lung sounds were collected at the eight locations (denoted by L1–L8) indicated in Fig 2A. The auscultation locations are described in detail in the caption of Fig 2. The two devices had a sampling rate of 4,000 Hz and a bit depth of 16 bits. The audio files were recorded in the WAVE (.wav) format.

All lung sounds in the TSECC database were collected using the Littmann 3200 device only, where 15.8-s recordings were obtained sequentially from L1 to L8 (Fig 2B; Littmann 3200).

**Table 1. Demographic data of patients.**

| | Subjects from RCW/RCC | Subjects in TSECC Database |
|---|---|---|
| Number (n) | 18 | 261 |
| Gender (M/F) | 11/7 | NA |
| Age | 67.5 (36.7, 98.3) | NA |
| Height (cm) | 163.6 (147.2, 180.0) | NA |
| Weight (kg) | 62.1 (38.2, 86.1) | NA |
| BMI (kg/m$^2$) | 23.1 (15.6, 30.7) | NA |
| Respiratory Diseases | | |
| Acute respiratory failure | 4 (22.2%) | NA |
| Chronic respiratory failure | 8 (44.4%) | NA |
| Acute exacerbation of chronic Obstructive pulmonary disease | 1 (5.6%) | NA |
| Chronic obstructive pulmonary disease | 2 (11.1%) | NA |
| Pneumonia | 4 (22.2%) | NA |
| Acute respiratory distress syndrome | 1 (5.6%) | NA |
| Emphysema | 1 (5.6%) | NA |
| Comorbidity | | |
| Chronic kidney disease | 1 (5.6%) | NA |
| Acute kidney injury | 3 (16.7%) | NA |
| Chronic heart failure | 2 (11.1%) | NA |
| Diabetes mellitus | 7 (38.9%) | NA |
| Hypertension | 6 (33.3%) | NA |
| Malignancy | 1 (5.6%) | NA |
| Arrythmia | 1 (5.6%) | NA |
| Cardiovascular disease | 1 (5.6%) | NA |

BMI: body mass index, RCW: respiratory care ward, RCC: respiratory care center, TSECC: Taiwan Smart Emergency and Critical Care. The mean values of the age, height, weight, and BMI are presented, with the corresponding 95% confidence interval in parentheses.

One round of recording with the Littmann 3200 device entails a recording of lung sounds from L1 to L8. The TSECC database was composed of data obtained from one to three rounds of recording with the Littmann 3200 device for each patient.

We recorded the lung sounds of the 18 RCW/RCC residents by using both the Littmann 3200 device and the HF-Type-1 device. The Littmann 3200 recording protocol was in accordance with that used in the TSECC database, except that data from four to five rounds of lung sound recording were collected instead. The HF-Type-1 device was used to record breath sounds at L1, L2, L4, L5, L6, and L8. One round of recording with the HF-Type-1 device entails a synchronous and continuous recording of breath sounds for 30 min (Fig 2B; HF-Type-1). However, the recording with the HF-Type-1 device was occasionally interrupted; in this case, the recording duration was <30 min.

Voluntary deep breathing was not mandated during the recording of lung sounds. The statistics of the recordings are listed in Table 2.

## Audio file truncation

In this study, the standard duration of an audio signal used for inhalation, exhalation, and adventitious sound detection was 15 s. This duration was selected because a 15-s signal contains at least three complete breath cycles, which are adequate for a clinician to reach a clinical

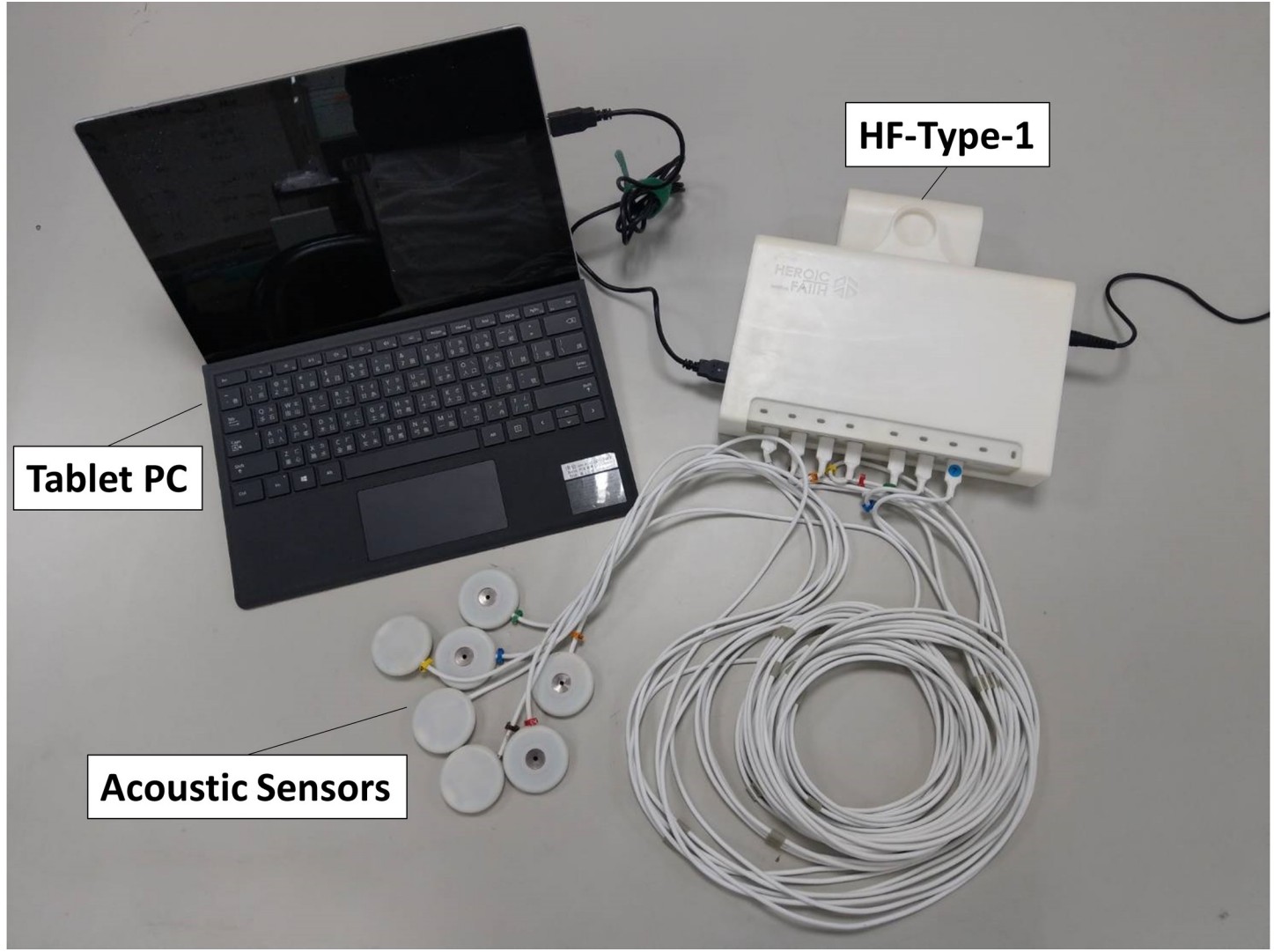

**Fig 1. Customized multichannel acoustic recording device (HF-Type-1) connected to a tablet.**

conclusion. Furthermore, a 15-s breath sound was used previously for verification and validation [66].

Because each audio file generated by the Littmann 3200 device had a length of 15.8 s, we cropped out the final 0.8-s signal from the files (Fig 2B; Littmann 3200). Moreover, we used only the first 15 s of each 2-min signal of the audio files (Fig 2B; HF-Type-1) generated by the HF-Type-1 device. Table 2 presents the number of truncated 15-s recordings and the total duration.

## Data labeling

Because the data in the TSECC database contains only classification labels indicating whether a CAS or DAS exists in a recording, we attempted to label the start and end time of all the events. Two board-certified respiratory therapists (NJL and YLW) and one board-certified nurse (WLT), with 8, 3, and 13 years of clinical experience, respectively, were recruited to label the start and end points of inhalation (I), exhalation (E), wheeze (W), stridor (S), rhonchus

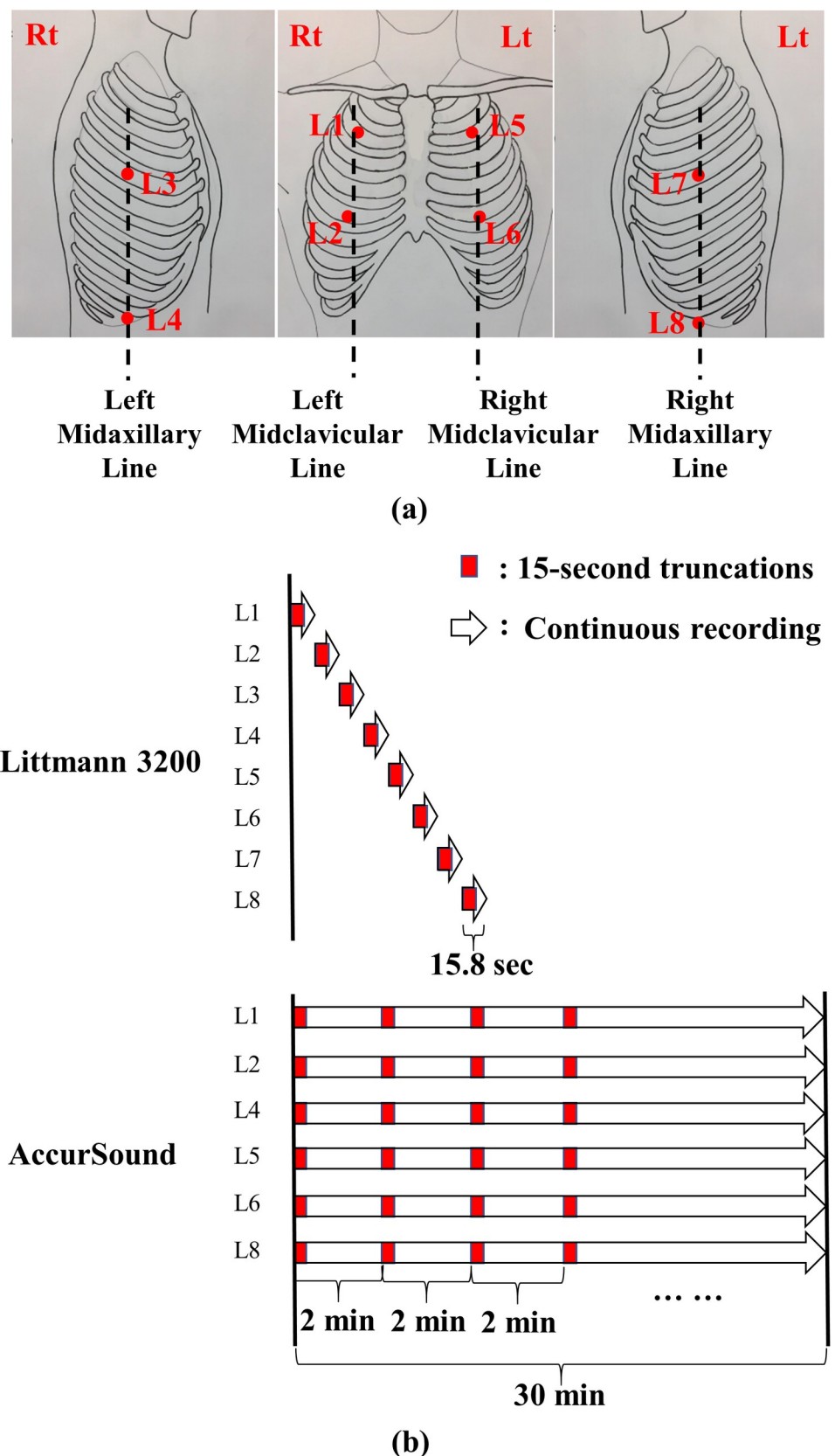

(a)

(b)

**Fig 2. Auscultation locations and lung sound recording protocol.** (a) Auscultation locations (L1–L8): L1: second intercostal space (ICS) on the right midclavicular line (MCL); L2: fifth ICS on the right MCL; L3: fourth ICS on the right midaxillary line (MAL); L4: tenth ICS on the right MAL; L5: second ICS on the left MCL; L6: fifth ICS on the left MCL; L7: fourth ICS on the left MAL; and L8: tenth ICS on the left MAL. (b) A standard round of breathing lung sound recording with Littmann 3200 and HF-Type-1 devices. The white arrows represent a continuous recording, and the small red blocks represent 15-s recordings. When the Littmann 3200 device was used, 15.8-s signals were recorded sequentially from L1 to L8. Subsequently, all recordings were truncated to 15 s. When the HF-Type-1 device was used, sounds at L1, L2, L4, L5, L6, and L8 were recorded simultaneously. Subsequently, each 2-min signal was truncated to generate new 15-s audio files.

(R), and DAS (D) events in the recordings. They labeled the sound events by listening to the recorded breath sounds while simultaneously observing the corresponding patterns on a spectrogram by using customized labeling software [67]. The labelers were asked not to label sound events if they could not clearly identify the corresponding sound or if an incomplete event at the beginning or end of an audio file caused difficulty in identification. BFH held regular meetings to ensure that the labelers had good agreement on labeling criteria based on a few samples by judging the mean pseudo-κ value [26]. When developing artificial intelligence (AI) detection models, we combined the W, S, and R labels to form CAS labels. Moreover, the D labels comprised only crackles, which were not differentiated into coarse or fine crackles. The labelers were asked to label the period containing crackles but not a single explosive sound (generally less than 25 ms) of a crackle. Each recording was annotated by only one labeler; thus, the labels did not represent perfect ground truth. However, we used the labels as ground-truth labels for model training, validation, and testing. The statistics of the labels are listed in Table 2.

**Table 2. Statistics of recordings and labels of HF_Lung_V1 database.**

|  | Littmann 3200 | HF-Type-1 | Total |
|---|---|---|---|
| Subjects |  |  |  |
| n | 261 | 18 | 261 |
| Recordings |  |  |  |
| Filename prefix | steth_ | trunc_ | NA |
| Rounds of recording | 748 | 70 | NA |
| No of 15-sec recordings | 4504 | 5261 | 9765 |
| Total duration (min) | 1126 | 1315.25 | 2441.25 |
| Labels |  |  |  |
| No of I | 16535 | 17560 | 34095 |
| Total duration of I (min) | 257.17 | 271.02 | 528.19 |
| Mean duration of I (s) | 0.93 | 0.93 | 0.93 |
| No of E | 9107 | 9242 | 18349 |
| Total duration of E (min) | 160.25 | 132.60 | 292.85 |
| Mean duration of E (s) | 1.06 | 0.86 | 0.96 |
| No of C/W/S/R | 6984/3974/152/2858 | 6899/4483/534/1882 | 13883/8457/686/4740 |
| Total duration of C/W/S/R (min) | 105.90/63.92/1.94/40.04 | 85.26/55.80/7.52/21.94 | 191.16/119.73/9.46/61.98 |
| Mean duration of C/W/S/R (s) | 0.91/0.97/0.76/0.84 | 0.74/0.75/0.85/0.70 | 0.83/0.85/0.83/0.78 |
| No of D | 7266 | 8340 | 15606 |
| Total duration of D (min) | 111.75 | 55.80 | 230.87 |
| Mean duration of D (s) | 0.92 | 0.87 | 0.89 |

I: inhalation, E: exhalation, W: wheeze, S: stridor, R: rhonchus, C: continuous adventitious sound, D: discontinuous adventitious sound. W, S, and R were combined to form C.

## Inhalation, exhalation, CAS, and DAS detection

### Framework

The inhalation, exhalation, CAS, and DAS detection framework developed in this study is displayed in Fig 3. The prominent advantage of the research framework is its modular design. Specifically, each unit of the framework can be tested separately, and the algorithms in different parts of the framework can be modified to achieve optimal overall performance. Moreover, the output of some blocks can be used for multiple purposes. For instance, the spectrogram generated by the preprocessing block can be used as the input of a model or for visualization in the user interface for real-time monitoring.

The framework comprises three parts: preprocessing, deep learning-based modeling, and postprocessing. The preprocessing part involves signal processing and feature engineering techniques. The deep learning-based modeling part entails the use of a well-designed neural network for obtaining a sequence of classification predictions rather than a single prediction. The postprocessing part involves merging the segment prediction results and eliminating the burst event.

### Preprocessing

We processed the lung sound recordings at a sampling frequency of 4 kHz. First, to eliminate the 60-Hz electrical interference and a part of the heart sound noise, we applied a high-pass filter to the recordings by setting a filter order of 10 and a cut-off frequency of 80 Hz. The filtered

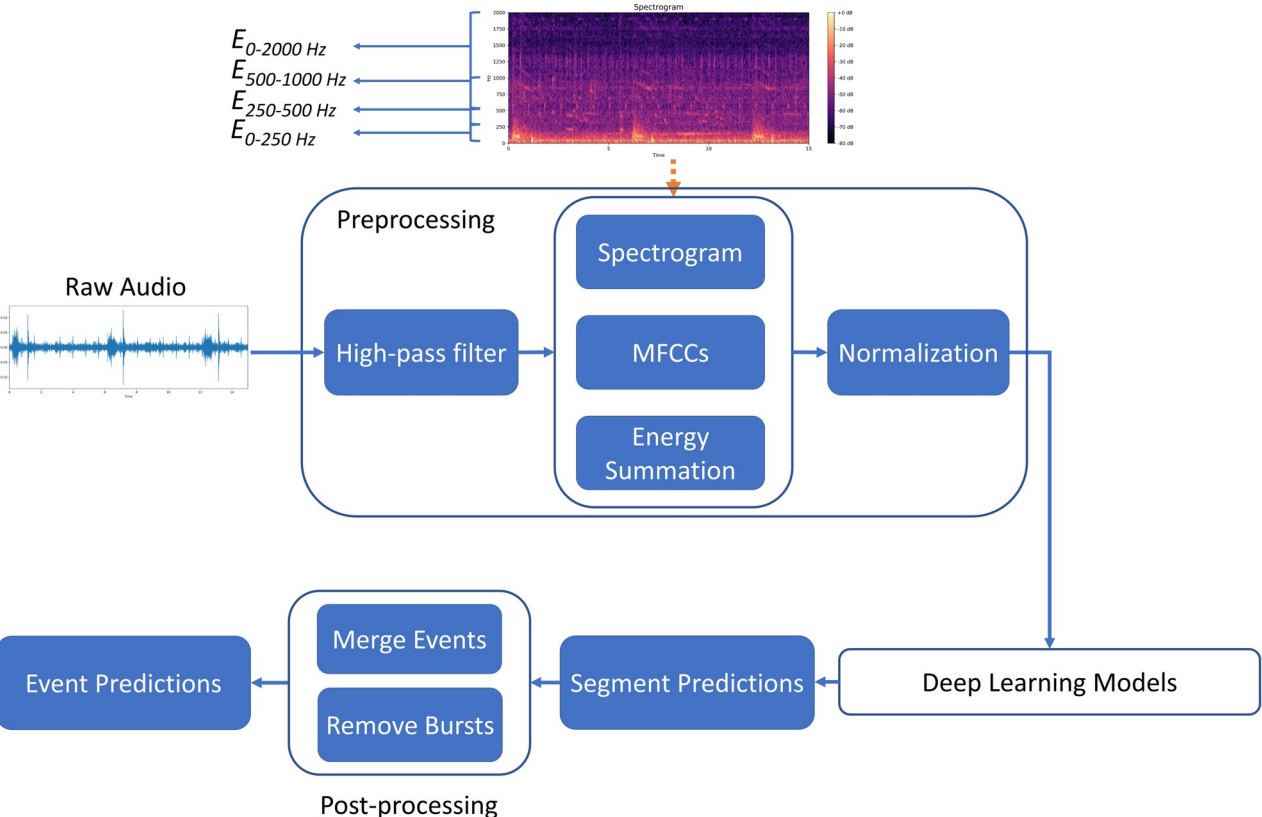

**Fig 3. Pipeline of detection framework.**

signals were then processed using STFT [25, 54, 68]. In the STFT, we set a Hanning window size of 256 and hop length of 64; no additional zero-padding was applied. Thus, a 15-s sound signal could be transformed into a corresponding spectrogram with a size of $938 \times 129$. To obtain the spectral information regarding the lung sounds, we extracted the following features [25, 54]:

- Spectrogram: We extracted 129-bin log-magnitude spectrograms.

- MFCCs [69]: We extracted 20 static coefficients, 20 delta coefficients ($\Delta$), and 20 acceleration coefficients ($\Delta^2$). We used 40 mel bands within a frequency range of 0–4,000 Hz. The frame width used to calculate the delta and acceleration coefficients was set to 9, which resulted in a 60-bin vector per frame.

- Energy summation: We computed the energy summation of four frequency bands, namely 0–250, 250–500, 500–1,000, and 0–2,000 Hz, and obtained four values per time frame.

After extracting the aforementioned features, we concatenated them to form a $938 \times 193$ feature matrix. Subsequently, we conducted min–max normalization on each feature. The values of the normalized features ranged between 0 and 1.

## Deep learning models

We investigated the performance of eight RNN-based models, namely LSTM, GRU, BiLSTM, BiGRU, CNN-LSTM, CNN-GRU, CNN-BiLSTM, and CNN-BiGRU, in terms of inhalation, exhalation, and adventitious sound detection. Fig 4 illustrates the detailed model structures. The LSTM [30] and GRU [31] models are able to use the past information to make a prediction at the present. However, bidirectional RNN models [32, 33], such as the BiLSTM and BiGRU models, can process the information not only from the past but also from the future with two separate hidden layers. The preceding CNN layers in the CNN-LSTM, CNN-BRU, CNN-BiLSTM, and CNN-BiGRU models can help extract abstract features first and then feed them as input to the following RNN layers [38]. Time distributed fully connected layers are added behind the LSTM, GRU, BiLSTM and BiGRU layers to further process the information and give a final prediction vector. The outputs of the LSTM, GRU, BiLSTM, and BiGRU models were $938 \times 1$ vectors, and those of the CNN-LSTM, CNN-GRU, CNN-BiLSTM, and CNN-BiGRU models were $469 \times 1$ vectors. An element in these vectors was set to 1 if an inhalation, exhalation, CAS, or DAS occurred within a time segment in which the output value passed the thresholding criterion; otherwise, the element was set to 0.

For a fairer comparison of the performance of the unidirectional and bidirectional models, we trained additional simplified (SIMP) BiLSTM, SIMP BiGRU, SIMP CNN-BiLSTM, and SIMP CNN-BiGRU models by adjusting the number of trainable parameters. Fig 5 illustrates the detailed architectures of the simplified bidirectional models. Parameter adjustment was conducted by halving the number of cells of the BiLSTM and BiGRU layers from 256 to 128.

We used Adam as the optimizer in the benchmark model, and we set the initial learning rate to 0.0001 with a step decay (0.2×) when the validation loss did not decrease for 10 epochs. The learning process stopped when no improvement occurred over 50 consecutive epochs.

## Postprocessing

The prediction vectors obtained using the adopted models can be further processed for different purposes. For example, we can transform the prediction result from frames to time for real-time monitoring. The breathing duration of most humans lies within a certain range; we

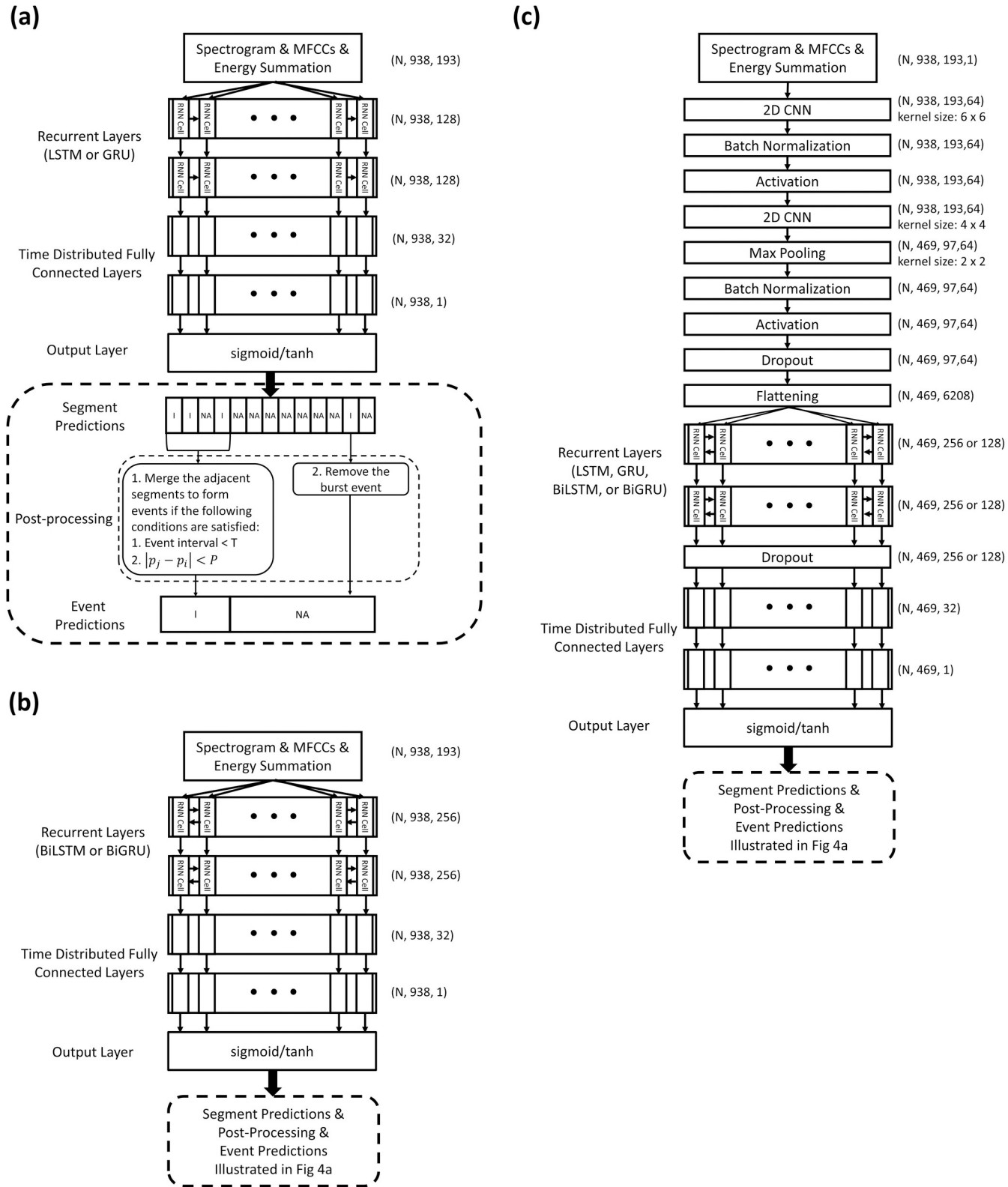

**Fig 4. Model architectures and postprocessing for inhalation, exhalation, CAS, and DAS segment and event detection.** (a) LSTM and GRU models; (b) BiLSTM and BiGRU models; and (c) CNN-LSTM, CNN-GRU, CNN-BiLSTM, and CNN-BiGRU models.

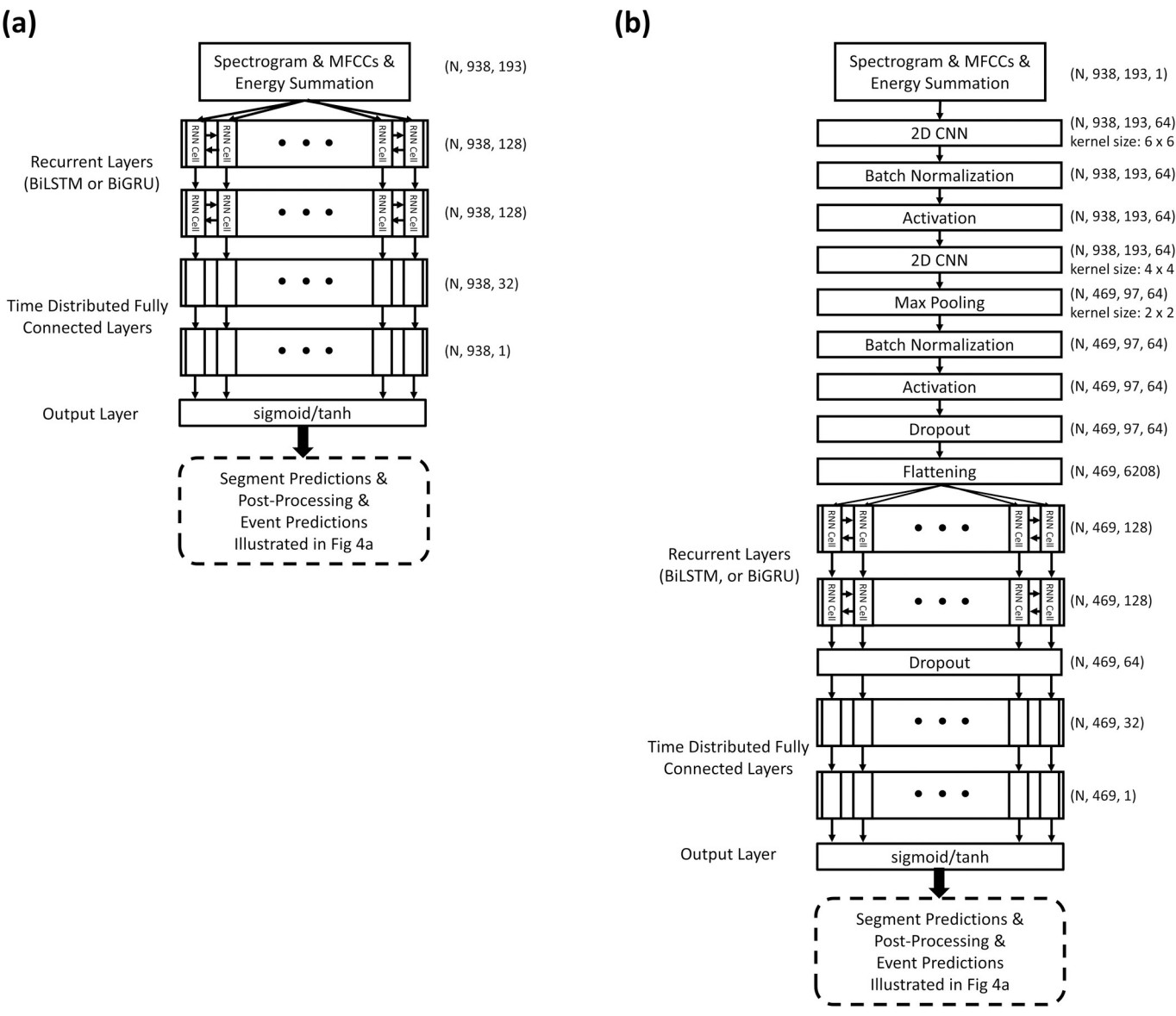

**Fig 5. Architectures of simplified bidirectional models.** (a) SIMP BiLSTM and SIMP BiGRU models; and (b) SIMP CNN-BiLSTM, and SIMP CNN-BiGRU models.

considered this fact in our study. Accordingly, when the prediction results obtained using the models indicated that two consecutive inhalation events occurred within a very small interval, we checked the continuity of these two events and decided whether to merge them, as illustrated in the bottom panel of Fig 4A. For example, when the interval between the $j$th and $i$th events was smaller than $T$ s, we computed the difference in frequency between their energy peaks ($|\boldsymbol{p_j}-\boldsymbol{p_i}|$). Subsequently, if the difference was below a given threshold $P$, the two events were merged into a single event. In the experiment, $T$ was set to 0.5 s, and $P$ was set to 25 Hz. After the merging process, we further assessed whether a burst event existed. If the duration of an event was shorter than 0.05 s, the event was deleted.

## Dataset arrangement and cross-validation

We adopted fivefold cross-validation in the training dataset to train and validate the models. More-over, we used an independent testing dataset to test the performance of the trained models. According to our preliminary experience, the acoustic patterns of the breath sounds collected from one patient at different auscultation locations or between short intervals had many similarities. To avoid potential data leakage caused by our methods of collecting and truncating the breath sound signals, we assigned all truncated recordings collected on the same day to only one of the training, validation, or testing datasets; this is because these recordings might have been collected from the same patient within a short period. The statistics of the datasets are listed in Table 3. We used only audio files containing CASs and DASs to train and test their corresponding detection models.

## Task definition and evaluation metrics

Pramono RXA, Bowyer S, and Rodriguez-Villegas E [4] clearly defined adventitious sounds classification and detection at the segment, event, and recording levels. In this study, we fol-lowed the definition and performed two tasks. The first task involved performing detection at the segment level. The acoustic signal of each lung sound recording was transformed into a spectrogram. The temporal resolution of the spectrogram depended on the window size and overlap ratio of the STFT. The aforementioned parameters were fixed such that each spectro-gram was a matrix of size $938 \times 129$. Thus, each recording contained 938 time segments (time frames), and each time segment was automatically labeled (Fig 6B) according to the ground-truth event labels (Fig 6A) assigned by the labelers. The output of the prediction process was a sequential prediction matrix (Fig 6C) of size $938 \times 1$ in the LSTM, GRU, BiLSTM, and BiGRU models and size $469 \times 1$ in the CNN-LSTM, CNN-GRU, CNN-BiLSTM, and CNN-BiGRU models. By comparing the sequential prediction with the ground-truth time segments, we could define true positive (TP; orange vertical bars in Fig 6D), true negative (TN; green vertical bars in Fig 6D), false positive (FP; black vertical bars in Fig 6D), and false negative (FN; yellow

**Table 3. Statistics of the datasets and labels of the HF_Lung_V1 database.**

|  | Training Dataset | Testing Dataset | Total |
|---|---|---|---|
| Recordings |  |  |  |
| No of 15-sec recordings | 7809 | 1956 | 9765 |
| Total duration (min) | 1952.25 | 489 | 2441.25 |
| Labels |  |  |  |
| No of I | 27223 | 6872 | 34095 |
| Total duration of I (min) | 422.17 | 105.97 | 528.14 |
| Mean duration of I (s) | 0.93 | 0.93 | 0.93 |
| No of E | 15601 | 2748 | 18349 |
| Total duration of E (min) | 248.05 | 44.81 | 292.85 |
| Mean duration of E (s) | 0.95 | 0.98 | 0.96 |
| No of C/W/S/R | 11464/7027/657/3780 | 2419/1430/29/960 | 13883/8457/686/4740 |
| Total duration of C/W/S/R (min) | 160.16/100.71/9.10/50.35 | 31.01/19.02/0.36/11.63 | 191.16/119.73/9.46/61.98 |
| Mean duration of C/W/S/R (s) | 0.84/0.86/0.83/0.80 | 0.77/0.80/0.74/0.73 | 0.83/0.85/0.83/0.78 |
| No of D | 13794 | 1812 | 15606 |
| Total duration of D (min) | 203.59 | 27.29 | 230.87 |
| Mean duration of D (s) | 0.89 | 0.90 | 0.89 |

I: inhalation, E: exhalation, W: wheeze, S: stridor, R: rhonchus, C: continuous adventitious sound, D: discontinuous adventitious sound. W, S, and R were combined to form C.

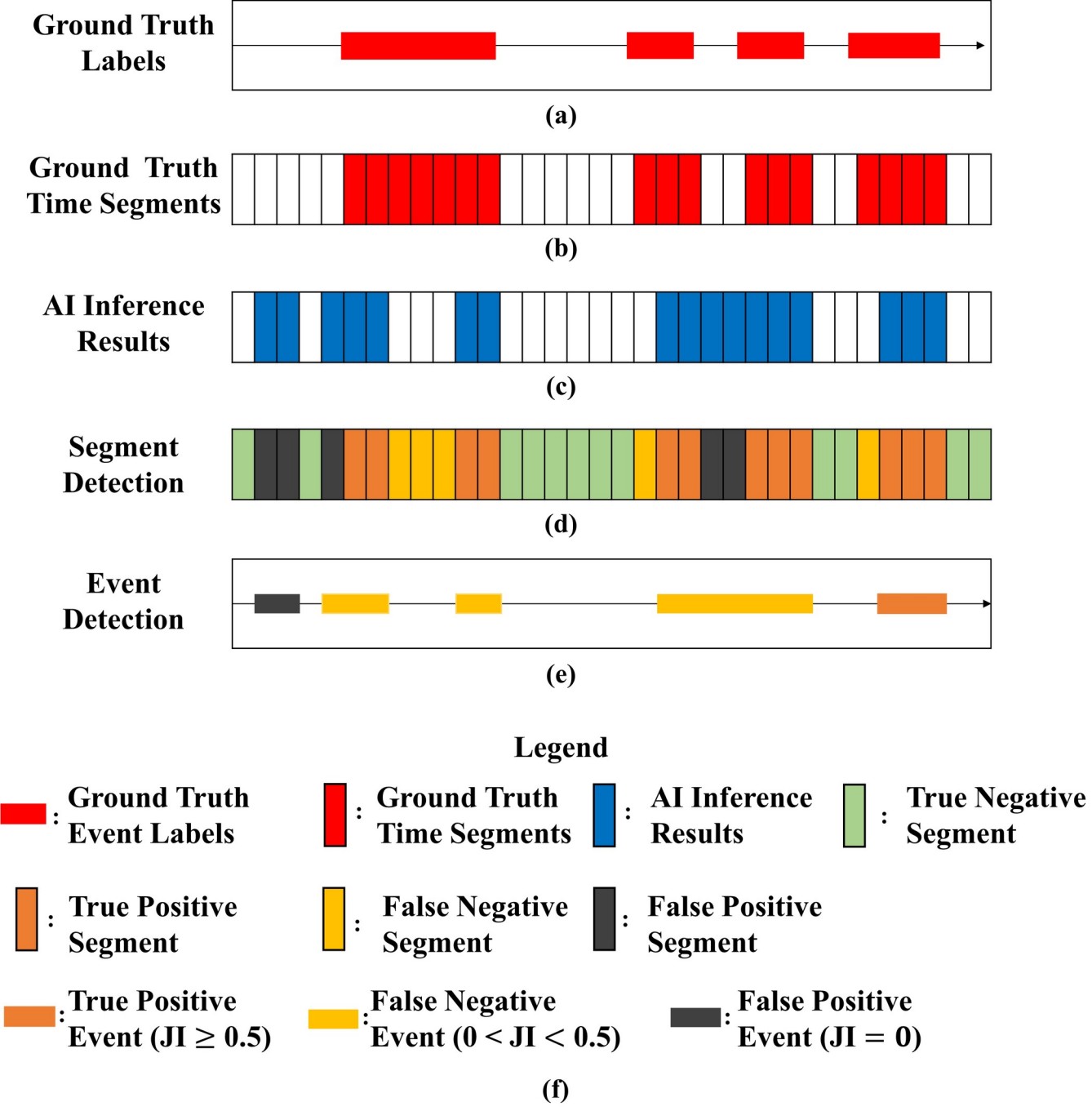

**Fig 6. Task definition and evaluation metrics.** (a) Ground-truth event labels, (b) ground-truth time segments, (c) AI inference results, (d) segment classification, (e) event detection, and (f) legend. JI: Jaccard index.

vertical bars in Fig 6D) time segments. Subsequently, the models' sensitivity and specificity in classifying the segments in each recording were computed.

The second task entailed event detection at the recording level. After completing the sequential prediction (Fig 6C), we assembled the time segments associated with the same label into a corresponding event (Fig 6E). We also derived the start and end times of each assembled event. The Jaccard index (JI; [26]) was used to determine whether an AI inference result

correctly matched the ground-truth event. For an assembled event to be designated as a TP event (orange horizontal bars in Fig 6E), the corresponding JI value must be greater than 0.5. If the JI was between 0 and 0.5, the assembled event was designated as an FN event (yellow horizontal bars in Fig 6E), and if it was 0, the assembled event was designated as an FP event (black horizontal bars in Fig 6E). A TN event cannot be defined in the task of event detection.

The performance of the models was evaluated using the *F1* score, and that of segment detection was evaluated using the receiver operating characteristic (ROC) curve and area under the ROC curve (AUC). In addition, the mean absolute percentage error (MAPE) of event detection was derived. The accuracy, positive predictive value (PPV), sensitivity, specificity, and *F1* score of the models are presented in the section of Supporting information.

## Hardware and software

We trained the baseline models on an Ubuntu 18.04 server provided by the National Center for High-Performance Computing in Taiwan [Taiwan Computing Cloud (TWCC)]. It was equipped with an Intel(R) Xeon(R) Gold 6154 @3.00 GHz CPU with 90 GB RAM. To manage the intensive computation involved in RNN training, we implemented the training module by using the TensorFlow 2.10, CUDA 10, and CuDNN 7 programs to run the NVIDIA Titan V100 card on the TWCC server for GPU acceleration.

## Results

### LSTM versus GRU models

Table 4 presents the *F1* scores used to compare the eight LSTM- and GRU-based models. When a CNN was not added, the GRU models outperformed the LSTM models by 0.7%–9.5% in terms of the *F1* scores. However, the CNN-GRU and CNN-BiGRU models did not outperform the CNN-LSTM and CNN-BiLSTM models in terms of the *F1* scores (and vice versa).

According to the ROC curves presented in Fig 7A–7D, the GRU-based models outperformed the LSTM-based models in all compared pairs, except for one pair, in terms of DAS segment detection (AUC of 0.891 for CNN-BiLSTM vs 0.889 for CNN-BiGRU).

### Unidirectional versus bidirectional models

As presented in Table 5, the bidirectional models outperformed their unidirectional counterparts in all the defined tasks by 0.4%–9.8% in terms of the *F1* scores, even when the bidirectional models had fewer trainable parameters after model adjustment.

**Table 4. Comparison of *F1* scores between LSTM-based models and GRU-based models.**

| Models | n of trainable parameters | Inhalation | | Exhalation | | CASs | | DASs | |
|---|---|---|---|---|---|---|---|---|---|
| | | *F1* score | | *F1* score | | *F1* score | | *F1* score | |
| | | Segment Detection | Event Detection | Segment Detection | Event Detection | Segment Detection | Event Detection | Segment Detection | Event Detection |
| LSTM | 300,609 | 73.9% | 76.1% | 51.8% | 57.0% | 15.1% | 12.2% | 62.6% | 59.1% |
| GRU | 227,265 | **76.2%** | **78.9%** | **59.8%** | **65.6%** | **24.6%** | **20.1%** | **65.9%** | **62.5%** |
| BiLSTM | 732,225 | 78.1% | 84.0% | 57.3% | 63.9% | 19.8% | 19.1% | 69.6% | 70.0% |
| BiGRU | 552,769 | **80.3%** | **86.2%** | **64.1%** | **70.9%** | **26.9%** | **25.6%** | **70.3%** | **71.4%** |
| CNN-LSTM | 3,448,513 | 77.6% | 81.1% | **57.7%** | **62.1%** | 45.3% | 42.5% | **68.8%** | 64.4% |
| CNN-GRU | 2,605,249 | **78.4%** | **82.0%** | 57.2% | 62.0% | **51.5%** | **49.8%** | 68.0% | **64.6%** |
| CNN-BiLSTM | 6,959,809 | **80.6%** | **86.3%** | 60.4% | 65.6% | 47.9% | 46.4% | **71.2%** | **70.8%** |
| CNN-BiGRU | 5,240,513 | **80.6%** | 86.2% | **62.2%** | **68.5%** | **53.3%** | **51.6%** | 70.6% | 70.0% |

The bold values indicate the higher *F1* score between the compared pairs of models.

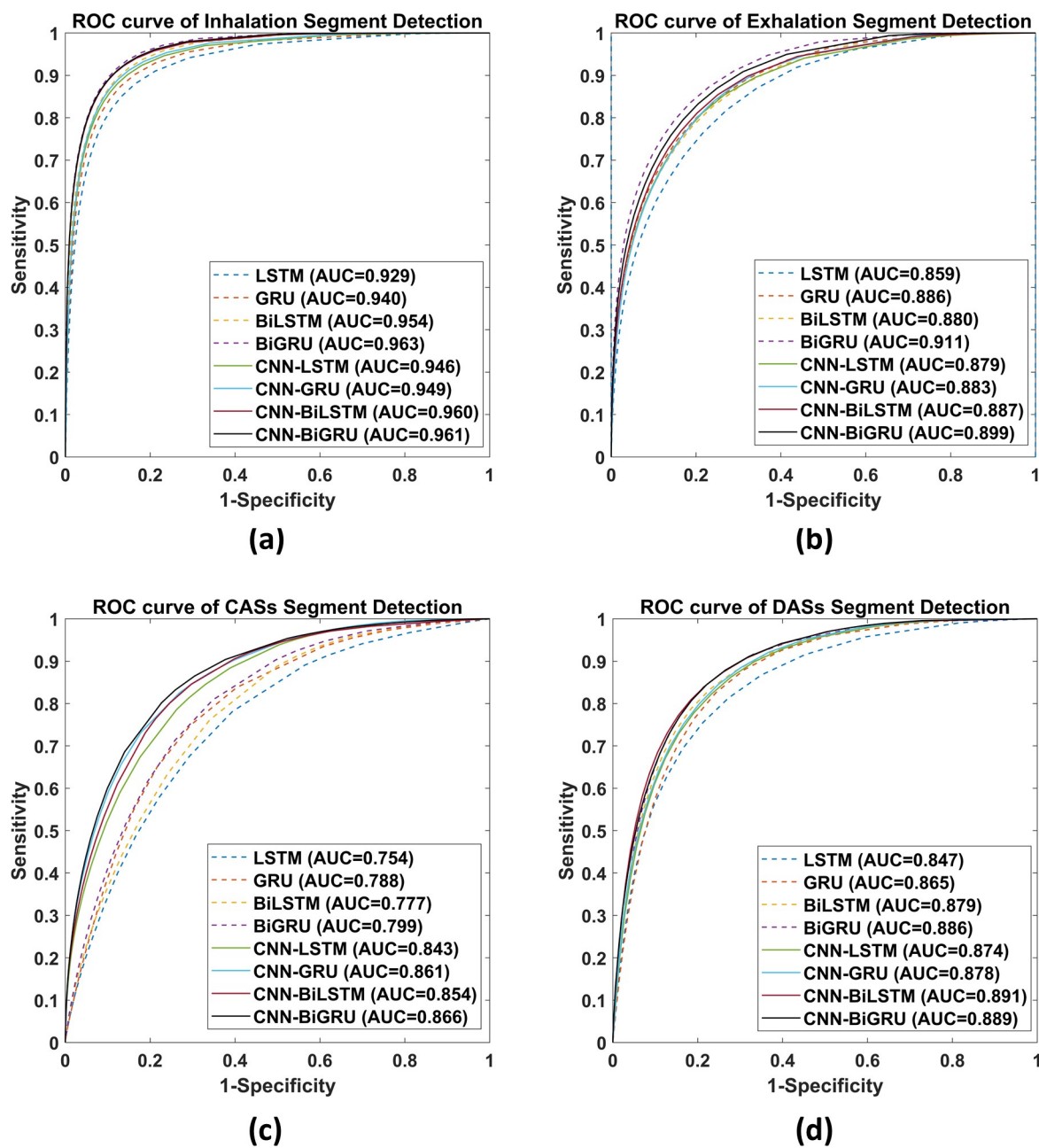

**Fig 7.** ROC curves for (a) inhalation, (b) exhalation, (c) CAS, and (d) DAS segment detection. The corresponding AUC values are presented.

### Models with CNN versus those without CNN

According to Table 6, the models with a CNN outperformed those without a CNN in 26 of the 32 compared pairs.

The models with a CNN exhibited higher AUC values than did those without a CNN (Fig 7A–7D), except that BiGRU had a higher AUC value than did CNN-BiGRU in terms of inhalation detection (0.963 vs 0.961), GRU had a higher AUC value than did CNN-GRU in terms of

**Table 5. Comparison of *F1* scores between the unidirectional and bidirectional models.**

| Models | n of trainable parameters | Inhalation | | Exhalation | | CASs | | DASs | |
|---|---|---|---|---|---|---|---|---|---|
| | | *F1* score | | *F1* score | | *F1* score | | *F1* score | |
| | | Segment Detection | Event Detection | Segment Detection | Event Detection | Segment Detection | Event Detection | Segment Detection | Event Detection |
| LSTM | 300,609 | 73.9% | 76.1% | 51.8% | 57.0% | 15.1% | 12.2% | 62.6% | 59.1% |
| SIMP BiLSTM | 235,073 | **77.8%** | **84.1%** | **55.8%** | **62.4%** | **19.8%** | **17.9%** | **68.8%** | **68.9%** |
| GRU | 227,265 | 76.2% | 78.9% | 59.8% | 65.6% | 24.6% | 20.1% | 65.9% | 62.5% |
| SIMP BiGRU | 178,113 | **80.1%** | **86.1%** | **63.7%** | **70.0%** | **25.0%** | **22.2%** | **70.3%** | **71.3%** |
| CNN-LSTM | 3,448,513 | 77.6% | 81.1% | 57.7% | 62.1% | 45.3% | 42.5% | 68.8% | 64.4% |
| SIMP CNN-BiLSTM | 3,382,977 | **80.0%** | **85.8%** | **60.4%** | **66.2%** | **50.8%** | **50.2%** | **70.2%** | **70.2%** |
| CNN-GRU | 2,605,249 | 78.4% | 82.0% | 57.2% | 62.0% | 51.5% | 49.8% | 68.0% | 64.6% |
| SIMP CNN-BiGRU | 2,556,097 | **80.1%** | **85.9%** | **62.4%** | **68.4%** | **52.6%** | **51.5%** | **69.9%** | **69.5%** |

The bold values indicate the higher *F1* score between the compared pairs of models. SIMP means the number of trainable parameters is adjusted.

exhalation detection (0.886 vs 0.883), and BiGRU had a higher AUC value than did CNN-BiGRU in terms of exhalation detection (0.911 vs 0.899).

Moreover, compared with the LSTM, GRU, BiLSTM, and BiGRU models, the CNN-LSTM, CNN-GRU, CNN-BiLSTM, and CNN-BiGRU models exhibited flatter and lower MAPE curves over a wide range of threshold values in all event detection tasks (Fig 8A–8D).

## Discussion

### Benchmark results

According to the *F1* scores presented in Table 4, among models without a CNN, the GRU and BiGRU models consistently outperformed the LSTM and BiLSTM models in all defined tasks. However, the GRU-based models did not have superior *F1* scores among models with a CNN. Regarding the ROC curves and AUC values (Fig 7A–7D), the GRU-based models consistently outperformed the other models in all but one task. Accordingly, we can conclude that GRU-based models perform slightly better than LSTM-based models in lung sound analysis.

**Table 6. Comparison of *F1* scores between models without and with a CNN.**

| Models | n of trainable parameters | Inhalation | | Exhalation | | CASs | | DASs | |
|---|---|---|---|---|---|---|---|---|---|
| | | *F1* score | | *F1* score | | *F1* score | | *F1* score | |
| | | Segment Detection | Event Detection | Segment Detection | Event Detection | Segment Detection | Event Detection | Segment Detection | Event Detection |
| LSTM | 300,609 | 73.9% | 76.1% | 51.8% | 57.0% | 15.10% | 12.20% | 62.60% | 59.10% |
| CNN-LSTM | 3,448,513 | **77.6%** | **81.1%** | **57.7%** | **62.1%** | **45.30%** | **42.50%** | **68.80%** | **64.40%** |
| BiLSTM | 732,225 | 76.2% | 78.9% | **59.8%** | **65.6%** | 19.80% | 17.90% | 68.80% | 68.90% |
| CNN-BiLSTM | 6,959,809 | **78.4%** | **82.0%** | 57.2% | 62.0% | **50.80%** | **50.20%** | **70.20%** | **70.20%** |
| GRU | 227,265 | 78.1% | 84.0% | 57.3% | 63.9% | 24.60% | 20.10% | 65.90% | 62.50% |
| CNN-GRU | 2,605,249 | **80.6%** | **86.3%** | **60.4%** | **65.6%** | **51.50%** | **49.80%** | **68.00%** | **64.60%** |
| BiGRU | 178,113 | 80.3% | **86.2%** | **64.1%** | **70.9%** | 25.00% | 22.20% | **70.30%** | **71.30%** |
| CNN-BiGRU | 2,556,097 | **80.6%** | **86.2%** | 62.2% | 68.5% | **52.60%** | **51.50%** | 69.90% | 69.50% |

The bold values indicate the higher *F1* score between the compared pairs of models.

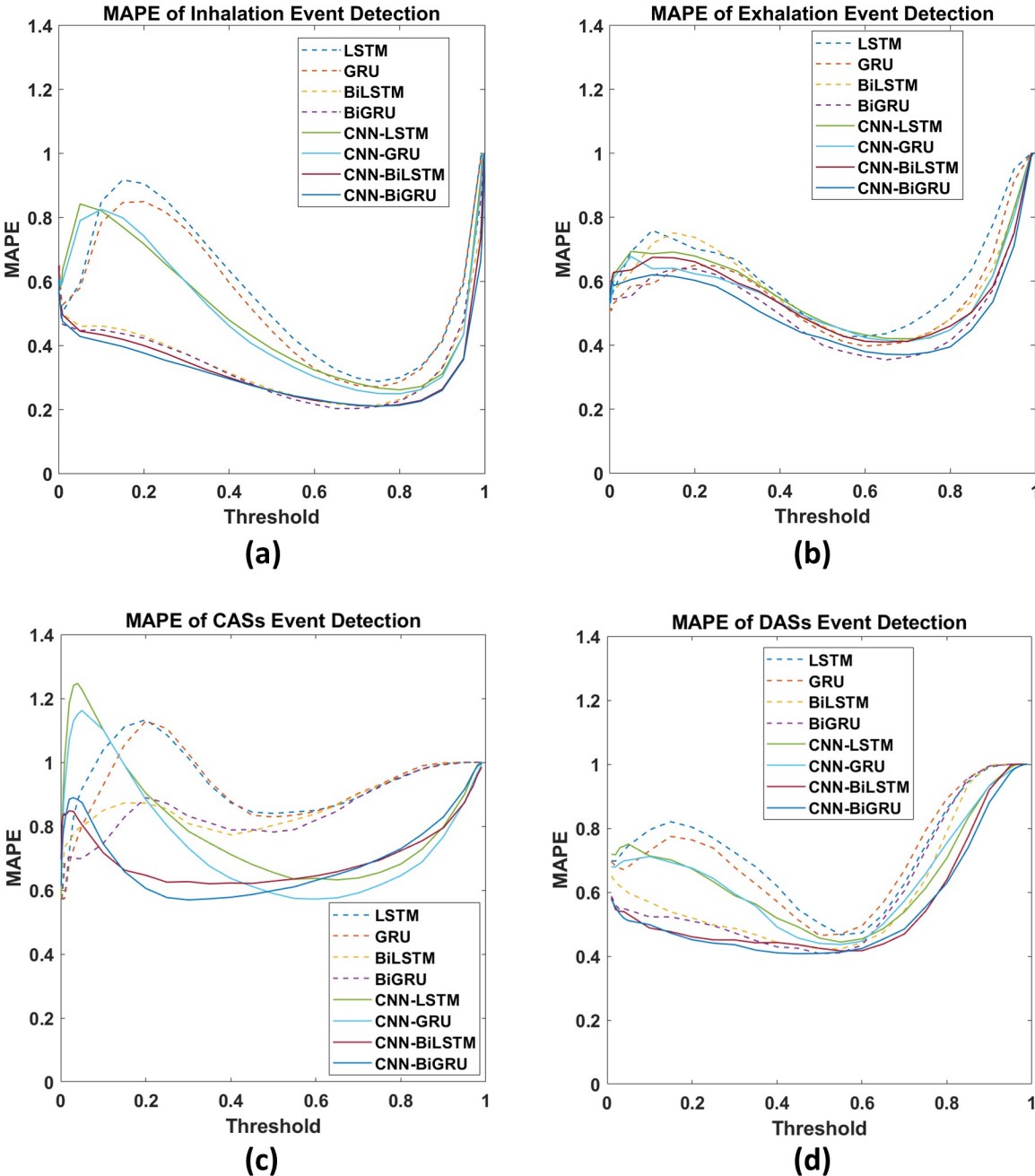

**Fig 8.** MAPE curves for (a) inhalation, (b) exhalation, (c) CAS, and (d) DAS event detection.

Previous studies have also compared LSTM- and GRU-based models [34, 70, 71]. Although a concrete conclusion cannot be drawn regarding whether LSTM-based models are superior to the GRU-based models (and vice versa), GRU-based models have been reported to outperform LSTM-based models in terms of computation time [34, 71].

As presented in Table 5, the bidirectional models outperformed their unidirectional counterparts in all defined tasks, a finding that is consistent with several previously obtained results [25, 32, 34, 36].

A CNN can facilitate the extraction of useful features and enhance the prediction accuracy of RNN-based models. The benefits engendered by a CNN are particularly vital in CAS detection. For the models with a CNN, the *F1* score improvement ranged from 26.0% to 30.3% and the AUC improvement ranged from 0.067 to 0.089 in the CAS detection tasks. Accordingly, we can infer that considerable information used in CAS detection resides in the local positional arrangement of the features. Thus, a two-dimensional CNN facilitates the extraction of the associated information. Notably, CNN-induced improvements in model performance in the inhalation, exhalation, and DAS detection tasks were not as high as those observed in the CAS detection tasks. The MAPE curves (Fig 8A–8D) reveal that a model with a CNN has more consistent predictions over various threshold values.

In our previous study [24], an attention-based encoder–decoder architecture based on ResNet and LSTM exhibited favorable performance in inhalation (*F1* score of 90.4%) and exhalation (*F1* score of 93.2%) segment detection tasks. However, the model was established on the basis of a very small dataset (489 recordings of 15-s-long lung sounds). Moreover, the model involves a complicated architecture; hence, it is impossible to implement real-time respiratory monitoring in devices with limited computing power, such as smartphones or medical-grade tablets.

Jácome et al. used a Faster R-CNN model for breath phase detection and achieved a sensitivity of 97.5% and specificity of 85% in inspiratory phase detection and a sensitivity of 95.5% and specificity of 82.5% in expiratory phase detection [26]. The evaluation method they used is similar but not the same as the one used for evaluating segment detection in the present study. Moreover, they reported the results after implementing post-processing; however, the results of our segment detection (S2 and S3 Tables) were derived before the post-processing was applied. Therefore, a fair comparison is not achievable.

Messner et al. [25] used the BiGRU model and one-dimensional labels (similar to those used in the present study) for breath phase and crackle detection. Their BiGRU model exhibited comparable performance to our models in terms of inhalation event detection (*F1* scores, 87.0% vs 86.2%) and in terms of DAS event detection (*F1* scores, 72.1% vs 71.4%). However, the performance of the BiGRU model differed considerably from that of our models in terms of exhalation detection (*F1* scores: 84.6% vs 70.9%). One of the reasons for this discrepancy is that Messner et al. established their ground-truth labels on the basis of the gold-standard signals of a pneumotachograph [25]. The second reason is that they may include pause phases into breath phases, but we focused on only labeling the events that can be heard. Another reason is that an exhalation label is not always available following an inhalation label in our data. Finally, we did not specifically control the sounds we recorded; for example, we did not ask patients to perform voluntary deep breathing or keep ambient noise down. The factors influencing the model performance are further discussed in the next section.

## Factors influencing model performance

The benchmark performance of the proposed models may have been influenced by the following factors: (1) unusual breathing patterns; (2) imbalanced data; (3) low signal-to-noise ratio (SNR); (4) noisy labels, including class and attribute noise, in the database; and (5) sound overlapping.

Fig 9A illustrates the general pattern of a breath cycle in the lung sounds when the ratio of inhalation to exhalation durations is approximately 2:1 and an expiratory pause is noted [3, 4]. However, in our recorded lung sound, an exhalation was sometimes not heard (Fig 9B). Moreover, because we did not ask the subjects to breathe voluntarily when recording the sound, many unusual breath patterns might have been recorded, such as patterns caused by shallow

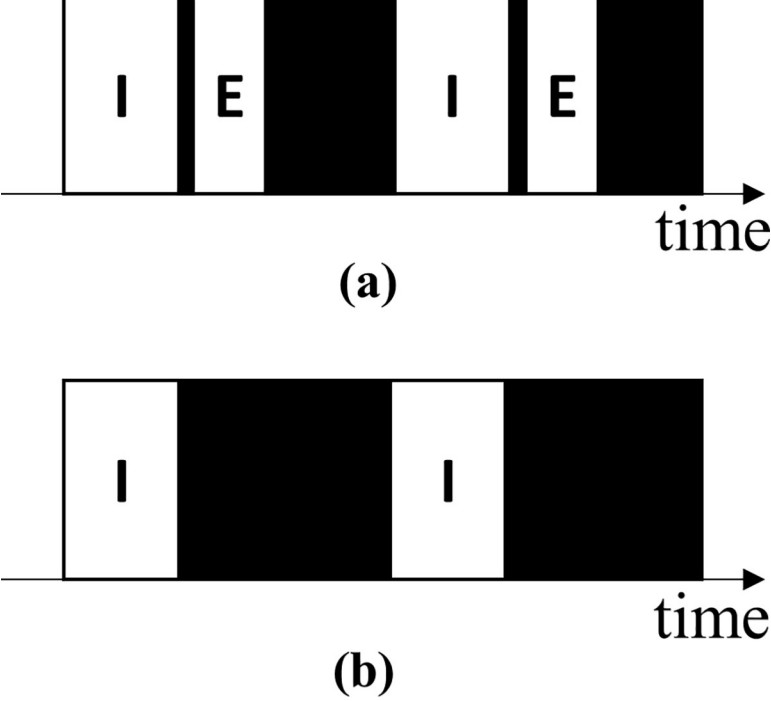

**Fig 9. Patterns of normal breathing lung sounds.** (a) General lung sound patterns and (b) general lung sound patterns with unidentifiable exhalations. "I" represents an identifiable inhalation event, "E" represents an identifiable exhalation event, and the black areas represent pause phases.

breathing, fast breathing, and apnea as well as those caused by double triggering of the ventilator [72] and air trapping [73, 74]. These unusual breathing patterns might confuse the labeling and learning processes and result in poor testing results.

The developed database contains imbalanced numbers of inhalation and exhalation labels (34,095 and 18,349, respectively) because not every exhalation was heard and labeled. In addition, the proposed models may possess the capability of learning the rhythmic rise and fall of breathing signals but not the capability of learning acoustic or texture features that can distinguish an inhalation from an exhalation. This may thus explain the models' poor performance in exhalation detection. However, these models are suitable for respiratory rate estimation and apnea detection as long as appropriate inhalation detection is achieved. Furthermore, for all labels, the summation of the event duration was smaller than that of the background signal duration (these factors had a ratio of approximately 1:2.5 to 1:7). The aforementioned phenomenon can be regarded as foreground–background class imbalance [75] and will be addressed in future studies.

Most of the sounds in the established database were not recorded during the patients performed deep breathing; thus, the signal quality was not maximized. However, training models with such nonoptimal data increase their adaptability to real-world scenarios. Moreover, the SNR may be reduced by noise, such as human voices; music; sounds from bedside monitors, televisions, air conditioners, fans, and radios; sounds generated by mechanical ventilators; electrical noise generated by touching or moving the parts of acoustic sensors; and friction sounds generated by the rubbing of two surfaces together (e.g., rubbing clothes with the skin). A poor SNR of audio signals can lead to difficulties in labeling and prediction tasks. The features of some noise types are considerably similar to those of adventitious sounds. The poor

performance of the proposed models in CAS detection can be partly attributed to the noisy environment in which the lung sounds were recorded. In particular, the sounds generated by ventilators caused numerous FP events in the CAS detection tasks. Thus, additional effort is required to develop a superior preprocessing algorithm that can filter out influential noise or to identify a strategy to ensure that models focus on learning the correct CAS features. Furthermore, the integration of active noise-canceling technology [76] or noise suppression technology [77] into respiratory sound monitors can help reduce the noise from auscultatory signals.

The sound recordings in the HF_Lung_V1 database were labeled by only one labeler; thus, some noisy labels, including class and attribute noise, may exist in the database [78]. These noisy labels are attributable to (1) the different hearing abilities of the labeler, which can cause differences in the labeled duration; (2) the absence of clear criteria for differentiating between target and confusing events; (3) individual human errors; (4) tendency to not label events located close to the beginning and end of a recording; and (5) confusion caused by unusual breath patterns and poor SNRs. However, deep learning models exhibit high robustness to noisy labels [79]. Accordingly, we are currently working toward establishing better ground-truth labels.

Breathing generates CASs and DASs under abnormal respiratory conditions. This means that the breathing sound, CAS, and DAS might overlap with one another during the same period. This sound overlapping, along with the data imbalance, makes the CAS and DAS detection models learn to read the rise and fall of the breathing energy and falsely identify an inhalation or exhalation as CAS or DAS, respectively. This FP detection was observed in our benchmark results. In the future, strategies must be adopted to address the problem of sound overlap.

## Conclusions

We established the largest open-access lung sound database, namely HF_Lung_V1 (https://gitlab.com/techsupportHF/HF_Lung_V1), that contains 9,765 audio files of lung sounds (each with a duration of 15 s), 34,095 inhalation labels, 18,349 exhalation labels, 13,883 CAS labels (comprising 8,457 wheeze labels, 686 stridor labels, and 4,740 rhonchus labels), and 15,606 DAS labels (all of which are crackles).

We also investigated the performance of eight RNN-based models in terms of inhalation, exhalation, CAS detection, and DAS detection in the HF_Lung_V1 database. We determined that the bidirectional models outperformed the unidirectional models in lung sound analysis. Furthermore, the addition of a CNN to these models further improved their performance.

Future studies can develop more accurate respiratory sound analysis models. First, highly accurate ground-truth labels should be established. Second, researchers should investigate the performance of state-of-the-art convolutional layers. Third, the advantage of using CNN variants can be maximized in lung sound analysis if the labels are expanded to two-dimensional bounding boxes on the spectrogram. Fourth, wavelet-based approaches, empirical mode decomposition, and other methods that can extract different features should be investigated [4, 80]. Finally, respiratory sound monitors should be equipped with the capability of tracheal breath sound analysis [76].

## Supporting information

**S1 Table. List of abbreviations.**
(DOCX)

**S2 Table. Accuracy, PPV, sensitivity, specificity, and *F1* scores of all models in inhalation detection.**
(DOCX)

**S3 Table. Accuracy, PPV, sensitivity, specificity, and *F1* scores of all models in exhalation detection.**
(DOCX)

**S4 Table. Accuracy, PPV, sensitivity, specificity, and *F1* scores of all models in CAS detection.**
(DOCX)

**S5 Table. Accuracy, PPV, sensitivity, specificity, and *F1* scores of all models in DAS detection.**
(DOCX)

## Acknowledgments

The authors thank the employees of Heroic Faith Medical Science Co. Ltd. who have ever partially contributed to developing the HF-Type-1 and establishing the HF_Lung_V1 database. This manuscript was edited by Wallace Academic Editing. The author would like to acknowledge the National Center for High-Performance Computing [Taiwan Computing Cloud] (TWCC) in providing computing resources. We also thank the All Vista Healthcare Center, Ministry of Science and Technology, Taiwan for the support.

## Author Contributions

**Data curation:** Yen-Chun Lai, Bi-Fang Hsu, Nian-Jhen Lin, Wan-Ling Tsai, Yi-Lin Wu, Tzu-Ling Tseng.

**Funding acquisition:** Fu-Shun Hsu.

**Investigation:** Shang-Ran Huang, Chien-Wen Huang.

**Methodology:** Shang-Ran Huang, Chien-Wen Huang, Yuan-Ren Cheng, Chun-Chieh Chen.

**Resources:** Jack Hsiao, Chung-Wei Chen.

**Supervision:** Fu-Shun Hsu, Yuan-Ren Cheng, Chun-Chieh Chen, Yen-Chun Lai, Feipei Lai.

**Writing – original draft:** Fu-Shun Hsu, Shang-Ran Huang, Chien-Wen Huang, Chao-Jung Huang.

**Writing – review & editing:** Li-Chin Chen, Ching-Ting Tseng, Yi-Tsun Chen, Feipei Lai.

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
