## [Decision Letter · Decision Letter 0]

21 Apr 2021

PONE-D-21-10240

Benchmarking of eight recurrent neural network variants for breath phase and adventitious sound detection on a self-developed open-access lung sound database—HF_Lung_V1

PLOS ONE

Dear Dr. Huang,

Thank you for submitting your manuscript to PLOS ONE. After careful consideration, we feel that it has merit but does not fully meet PLOS ONE’s publication criteria as it currently stands. Therefore, we invite you to submit a revised version of the manuscript that addresses the points raised during the review process.

Based on the comments received from the reviewers and my own observations, I recommend major revisions for the article.

We look forward to receiving your revised manuscript.

Kind regards,

Thippa Reddy Gadekallu

Academic Editor

PLOS ONE

Journal Requirements:

'This study was partially funded by the Raising Children Medical Foundation, Taiwan (http://http://www.raising.org.tw). The funders had no role in study design, data collection and analysis, decision to publish, or preparation of the manuscript.'

a. Please provide an amended statement that declares *all* the funding or sources of support (whether external or internal to your organization) received during this study, as detailed online in our guide for authors at http://journals.plos.org/plosone/s/submit-now

Please also include the statement “There was no additional external funding received for this study.” in your updated Funding Statement.

'The authors have declared that no competing interests exist.'

We note that one or more of the authors are employed by commercial companies: Heroic Faith Medical Science Co. Ltd , Avalanche Computing Inc and HCC Healthcare Group.

b. Please also provide an updated Competing Interests Statement declaring these commercial affiliations along with any other relevant declarations relating to employment, consultancy, patents, products in development, or marketed products, etc.  

Within your Competing Interests Statement, please confirm that these commercial affiliations do not alter your adherence to all PLOS ONE policies on sharing data and materials by including the following statement: "This does not alter our adherence to  PLOS ONE policies on sharing data and materials.” (as detailed online in our guide for authors http://journals.plos.org/plosone/s/competing-interests) . If this adherence statement is not accurate and  there are restrictions on sharing of data and/or materials, please state these. Please note that we cannot proceed with consideration of your article until this information has been declared.

Reviewers' comments:

Reviewer's Responses to Questions

**Comments to the Author**

1. Is the manuscript technically sound, and do the data support the conclusions?

Reviewer #1: No

Reviewer #2: Yes

Reviewer #3: Yes

2. Has the statistical analysis been performed appropriately and rigorously? 

Reviewer #1: Yes

Reviewer #2: Yes

Reviewer #3: Yes

3. Have the authors made all data underlying the findings in their manuscript fully available?

Reviewer #1: Yes

Reviewer #2: Yes

Reviewer #3: Yes

4. Is the manuscript presented in an intelligible fashion and written in standard English?

Reviewer #1: No

Reviewer #2: Yes

Reviewer #3: Yes

5. Review Comments to the Author

Reviewer #1: Dear authors, it was pleasure reading the paper titled "Benchmarking of eight recurrent neural network variants for breath phase and adventitious sound detection on a self-developed open-access lung sound database—HF_Lung_V1". I have some suggestions/comments for you to address. Please find the attached file for comments and suggestions.

Reviewer #2: The first objective of this paper is to establish a large and open-access respiratory sound database for training such an algorithm for the detection of breath phase and adventitious sounds, mainly focusing on lung sounds. The second objective is to conduct a benchmark test on the established lung sound database by using eight recurrent neural network-based models. (Eight RNN models namely LSTM, GRU, BiLSTM, BiGRU, CNN-LSTM, CNN-GRU, CNN-BiLSTM and CNN-BiLSTM).

Strengths:

- Clear description of the problematic (prolonged continuous monitoring of respiratory sound is almost impractical, the interpretation of auscultation results depends on the subjectivity of the practitioners, the importance of data size for creating robust and accurate DL models…).

-The results and the data are well presented.

Weaknesses:

The paper is well written, However, during the review process, I found few queries enlisted below that may be helpful to improve the quality of the paper with proper revision.

- Subsection “3.6 Task definition and evaluation metrics” line 293 [4] clearly —> need a proper citation.

- Line 433 “double triggering” —> “a double triggering”.

- It would be good to have a table that lists the abbreviations.

- The authors need to explain the eight recurrent neural network models.

- The title of subsection “3.2 Preprocessing”, “3.7 Hardware and software”, and “2.4 Data labeling” are badly placed in the text.

- The title of Table 1 is badly placed.

- The authors need to improve the quality of Figure 2, Figure 3, Figure 3, Figure 6, and Figure 7.

- The authors need to provide more details on the simplified version of the RNN-based models.

- “The filtered signals were then processed using the short-time Fourier transform (STFT)”, The authors need to explain the STFT method.

- The authors need to add a related work section to clearly position their work comparing to others.

Reviewer #3: 1. In Introduction section, the drawbacks of each conventional technique should be described clearly.

2. Introduction needs to explain the main contributions of the work more clearly.

3. The authors should emphasize the difference between other methods to clarify the position of this work further.

4. The Wide ranges of applications need to be addressed in Introductions

5. The objective of the research should be clearly defined in the last paragraph of the introduction section.

6. Fig 6 quality should be increased

7. The authors can cite the following references

Early detection of diabetic retinopathy using PCA-firefly based deep learning model

An ensemble based machine learning model for diabetic retinopathy classification

Deep learning and medical image processing for coronavirus (COVID-19) pandemic: A survey

6. PLOS authors have the option to publish the peer review history of their article (what does this mean?). If published, this will include your full peer review and any attached files.

Reviewer #1: No

Reviewer #2: No

Reviewer #3: No

---

## [Author Response · Author response to Decision Letter 0]

4 Jun 2021

Responses to the editor

Ans: We have checked the style guidelines. We have readjusted the font size and font weight of the headings and subheadings. We have removed the funding information from the Acknowledgments.

'This study was partially funded by the Raising Children Medical Foundation, Taiwan (http://http://www.raising.org.tw). The funders had no role in study design, data collection and analysis, decision to publish, or preparation of the manuscript.'

a. Please provide an amended statement that declares *all* the funding or sources of support (whether external or internal to your organization) received during this study, as detailed online in our guide for authors at http://journals.plos.org/plosone/s/submit-now

Please also include the statement “There was no additional external funding received for this study.” in your updated Funding Statement.

Ans: We have amended the “Funding Statement” as follows: “Raising Children Medical Foundation, Taiwan, fully funded all of the lung sound collection and contributed the recordings to Taiwan Society of Emergency and Critical Care Medicine. The Heroic Faith Medical Science Co. Ltd, Taipei, Taiwan, freely provided the lung sound recording device (HF-Type-1) for the study and fully sponsored the data labeling and deep learning model training. There was no additional external funding received for this study.

This statement is also included in the cover letter.

'The authors have declared that no competing interests exist.'

We note that one or more of the authors are employed by commercial companies: Heroic Faith Medical Science Co. Ltd , Avalanche Computing Inc and HCC Healthcare Group.

b. Please also provide an updated Competing Interests Statement declaring these commercial affiliations along with any other relevant declarations relating to employment, consultancy, patents, products in development, or marketed products, etc. 

Within your Competing Interests Statement, please confirm that these commercial affiliations do not alter your adherence to all PLOS ONE policies on sharing data and materials by including the following statement: "This does not alter our adherence to PLOS ONE policies on sharing data and materials.” (as detailed online in our guide for authors http://journals.plos.org/plosone/s/competing-interests) . If this adherence statement is not accurate and there are restrictions on sharing of data and/or materials, please state these. Please note that we cannot proceed with consideration of your article until this information has been declared.

Ans: We have revised the “Competing Interests” as follows: “FSH, SRH, YRC, YCL, BFH, YLW, TLT and CTT are full-time employees and CJH, NJL, WLT and YTC are part-time employees of Heroic Faith Medical Science Co. Ltd. CWH and CHC are with Avalanche Computing Inc., whom Heroic Faith Medical Science Co. Ltd. commissioned to train the deep learning models. This does not alter our adherence to PLOS ONE policies on sharing data and materials.

This statement is also included in the cover letter.

Ans: We have added the captions of the Supporting information files at the end of the manuscript. The in-text citations are updated accordingly.

 

Responses to the reviewers

Reviewer #1:

1. What does benchmarking means here? Are they combining all these 8 models to generate single model?

Ans: Benchmarking means a process of measuring the performance of the 8 models in breath phase and adventitious respiratory sound detection based on the newly established database. The importance of benchmarking is to provide a baseline reference for the future studies.

The answer to the second question is “No”. We reported the performance of all the 8 models in the lung sound analysis. Because we also tried to compare the performance between the 8 models in this study. 

2. Why RNN? What’s so special in RNN that is not covered by other variants of deep learning as well as the conventional learning?

Ans: Because the auscultatory signal is temporal and the labels are one dimensional. The action of breathing is periodic and rhythmic, and the mechanisms of inhalation, exhalation and adventitious sound generation are temporally associated. RNN is widely accepted to analyze temporal signals. 

We have experimented some models using only CNN variants. However, we found RNN can help refine the prediction results given by CNN and get a more stable prediction results when the dataset is still not large enough.

A new paragraph is also added to explain why we used deep learning instead of traditional machine learning (line 73–79).

3. What does “S” means? Seconds?

Ans: Yes, “s” means second. According to some writing guidelines, such as the one provided by IEEE (https://www.ewh.ieee.org/soc/ias/pub-dept/abbreviation.pdf), “s“ is a preferred abbreviation for second in SI unit. 

4. The major point is: What is new in all these the authors proposed?

Ans: The major aims of the present study are to establish the largest open-access lung sound database and to report the benchmark tests using 8 different RNN-based models. In addition, the performance between the 8 RNN-based models is also compared. The study aims are summarized from line 123 to line 130 in the revised manuscript.

We have also revised the sentences in the sections of “Introduction” and “Related work” and positioned our work compared to the others more clearly. We have explained that so far only few studies have reported the results of sound detection at the recording level in respiratory sound analysis based on private datasets (line 92–98 and line 133–150). Therefore, without our benchmarking results and the open-access database, there are no baseline reference to be compared to in the future studies. We expect the benchmark results and the open-access database presented in this paper can greatly contribute to the development of respiratory sound analysis methods in the future.

5. Which variant of CNN? There are many variants of CNN has been proposed, for instance, 3-D, 2-D, hybrid CNN, etc. Your are requested to check the other CNN models of CNN to bring some novelty in your work and the following repository may help you in your current and future studies https://github.com/mahmad00

Ans: The CNN in this study means a simple 2D CNN with two convolutional layers. The detailed architectures of the networks can be found in Fig 4 and Fig 5. 

We believe that advanced CNN variants could have better performance in feature extraction, and actually we have already experimented some of them. However, the benchmark results presented in this paper is just to serve as the first anchor for the future studies. Additionally, we consider using the models in a smart device for real-time respiratory monitoring so computing cost is a concern when using some complicated CNN structures or deeper CNN layers. Therefore, we started the research from some combinations of simple CNN and RNN layers. 

It is mentioned in the section of “Conclusion” that the state-of-the-art CNN layers can be investigated in the future (line 551–552) and the advantage of using region-based CNN variants can be maximized in lung sound analysis if the labels are expanded to two-dimensional bounding boxes on the spectrogram (line 552– 554). 

Thank you for providing the materials for us to research in the future. We will keep finding the most efficient and best solution to automated respiratory sound analysis.

 

Reviewer #2: 

1. Subsection “3.6 Task definition and evaluation metrics” line 293 [4] clearly —> need a proper citation.

Ans: We have revised the sentence. It is now in the line 347 in the revised manuscript.

2. Line 433 “double triggering” —> “a double triggering”.

Ans: Thanks for the suggestion. However, after googling “a double triggering”, we found double triggering is used as an uncountable term in most search results. Hence, we decided not to add an “a” in the text.

3. It would be good to have a table that lists the abbreviations.

Ans: Thanks for the suggestion. We have put a list of abbreviations in S1 Table 1 in the section of “Supplementary information”.

4. The authors need to explain the eight recurrent neural network models.

Ans: We have added some descriptions to briefly explain the eight recurrent neural network models (line 288–295) and references are cited.

5. The title of subsection “3.2 Preprocessing”, “3.7 Hardware and software”, and “2.4 Data labeling” are badly placed in the text.

Ans: Thank you for the kind reminder. We have adjusted the position. The final typesetting will be helped by the staff of PLOS ONE. 

6. The title of Table 1 is badly placed.

Ans: Thank you for the kind reminder. We have adjusted the position. The final typesetting will be helped by the staff of PLOS ONE. 

7. The authors need to improve the quality of Figure 2, Figure 3, Figure 3, Figure 6, and Figure 7.

Ans: The quality of these figures is readjusted and checked by PACE as suggested by the editor. However, the submission system automatically downgrades the quality for some reasons when generating the pdf file for reviewing. The quality of the figure looks fine if it is downloaded from the provided link. We will keep communicating with the staff of PLOS ONE to make sure the quality is good when the paper is published.

8. The authors need to provide more details on the simplified version of the RNN-based models.

Ans: We have added a new figure (Fig 5) to illustrate the structures of simplified convolutional RNN models. Corresponding explanation is also added from the line 307 to line 309 in the revised manuscript.

9. “The filtered signals were then processed using the short-time Fourier transform (STFT)”, The authors need to explain the STFT method.

Ans: We have cited a book (Cohen L. Time-frequency analysis: Prentice Hall PTR Englewood Cliffs, NJ; 1995) and two previous studies to help explain the STFT. The parameters used in STFT are clearly described from line 270 to line 272. 

10. The authors need to add a related work section to clearly position their work comparing to others.

Ans: We have reorganized the article and added a “Related work” section (line 133 to line 150). We clearly stated that so far only few other studies have reported the results of sound detection at the recording level in the respiratory sound analysis, and the dataset used in these studies are not open to the public. Hence, the benchmark results and the open-access database can be the foundation for future studies.

 

Reviewer #3: 

1. In Introduction section, the drawbacks of each conventional technique should be described clearly.

Ans: In the first paragraph of the “Introduction” section, the limitations of conventional auscultation are explained clearly (line 58–67).

We have also added descriptions (line 73–79) to explain the drawback of using conventional machine learning and the strength of using deep learning to research computerized respiratory sound analysis.

2. Introduction needs to explain the main contributions of the work more clearly.

Ans: We have revised the manuscript and explained the reasons why we conducted this study: (1) In regard to the amount of data used in the deep learning, the more the better (line 88–91), and (2) only few studies have reported the results of sound detection at the recording level in the respiratory sound analysis based on private datasets (line 92–101). We clearly stated the aims of the studies in the last paragraph of introduction section (line 122–130).

3. The authors should emphasize the difference between other methods to clarify the position of this work further.

Ans: We have added a new section “Related work” (line 132–150). In this section, we have clearly stated that most studies focused on only the sound classification tasks. Only few studies have reported the results of sound detection at the recording level in the respiratory sound analysis, albeit based on private datasets. 

4. The Wide ranges of applications need to be addressed in Introductions

Ans: We have added some descriptions to explain the application of computerized respiratory sound analysis (line 67–72). It can not only overcome the limitations of conventional auscultation but facilitate a tele-auscultation system.

5. The objective of the research should be clearly defined in the last paragraph of the introduction section.

Ans: We have revised the “Introduction: section to clarify why we conducted this study (line 84–101). The aims of the study have been summarized in the last paragraph of the introduction section (line 122–130).

6. Fig 6 quality should be increased

Ans: The quality of the figure is readjusted and checked by PACE as suggested by the editor. However, the submission system automatically downgrades the quality for some reasons when generating the pdf file for reviewing. The quality of the figure looks fine if it is downloaded from the provided link. We will keep communicating with the staff of PLOS ONE to make sure the quality is good when the paper is published.

7. The authors can cite the following references

Early detection of diabetic retinopathy using PCA-firefly based deep learning model

An ensemble based machine learning model for diabetic retinopathy classification

Deep learning and medical image processing for coronavirus (COVID-19) pandemic: A survey

Ans: We have added a new paragraph in the “Introduction” section to discuss the reasons of using deep learning (line 74–79). We mentioned that deep learning is widely used in combating COVID-19 (line 80–81), and we have cited the third reference. 

The other two papers are not closely related to this study. However, these references provide useful information for developing a better deep learning model for respiratory sound analysis. We could cite them in our future publications.

---

## [Decision Letter · Decision Letter 1]

21 Jun 2021

Benchmarking of eight recurrent neural network variants for breath phase and adventitious sound detection on a self-developed open-access lung sound database—HF_Lung_V1

PONE-D-21-10240R1

Dear Dr. Huang,

We’re pleased to inform you that your manuscript has been judged scientifically suitable for publication and will be formally accepted for publication once it meets all outstanding technical requirements.

Kind regards,

Thippa Reddy Gadekallu

Academic Editor

PLOS ONE

Additional Editor Comments (optional):

Reviewers' comments:

Reviewer's Responses to Questions

**Comments to the Author**

1. If the authors have adequately addressed your comments raised in a previous round of review and you feel that this manuscript is now acceptable for publication, you may indicate that here to bypass the “Comments to the Author” section, enter your conflict of interest statement in the “Confidential to Editor” section, and submit your "Accept" recommendation.

Reviewer #2: All comments have been addressed

Reviewer #3: All comments have been addressed

2. Is the manuscript technically sound, and do the data support the conclusions?

Reviewer #2: Yes

Reviewer #3: Yes

3. Has the statistical analysis been performed appropriately and rigorously? 

Reviewer #2: Yes

Reviewer #3: Yes

4. Have the authors made all data underlying the findings in their manuscript fully available?

Reviewer #2: Yes

Reviewer #3: Yes

5. Is the manuscript presented in an intelligible fashion and written in standard English?

Reviewer #2: Yes

Reviewer #3: Yes

6. Review Comments to the Author

Reviewer #2: The paper has been improved significantly.

In general, the paper is well sounded, and the comparative analysis of the eight recurrent neural network variants is interesting.

Reviewer #3: The authors have addressed all of my comments. The paper can can be accepted in the current format. Thank you

7. PLOS authors have the option to publish the peer review history of their article (what does this mean?). If published, this will include your full peer review and any attached files.

Reviewer #2: No

Reviewer #3: No

---

## [Editor Report · Acceptance letter]

23 Jun 2021

PONE-D-21-10240R1 

Benchmarking of eight recurrent neural network variants for breath phase and adventitious sound detection on a self-developed open-access lung sound database—HF_Lung_V1 

Dear Dr. Huang:

I'm pleased to inform you that your manuscript has been deemed suitable for publication in PLOS ONE. Congratulations! Your manuscript is now with our production department. 

Kind regards, 

on behalf of

Dr. Thippa Reddy Gadekallu 

Academic Editor

PLOS ONE